# Moderate-Intensity Exercise Improves Mesenteric Arterial Function in Male UC Davis Type-2 Diabetes Mellitus (UCD-T2DM) Rats: A Shift in the Relative Importance of Endothelium-Derived Relaxing Factors (EDRF)

**DOI:** 10.3390/biomedicines11041129

**Published:** 2023-04-08

**Authors:** Md Rahatullah Razan, Said Amissi, Rifat Ara Islam, James L. Graham, Kimber L. Stanhope, Peter J. Havel, Roshanak Rahimian

**Affiliations:** 1Department of Physiology and Pharmacology, Thomas J. Long School of Pharmacy, University of the Pacific, Stockton, CA 95211, USA; 2Department of Molecular Biosciences, School of Veterinary Medicine, University of California, Davis, CA 95616, USA; 3Department of Nutrition, University of California, Davis, CA 95616, USA

**Keywords:** type-2 diabetes (T2D), moderate-intensity exercise (MIE), endothelium-derived relaxing factors (EDRF), mesenteric artery, acetylcholine (ACh)

## Abstract

The beneficial cardiovascular effects of exercise are well documented, however the mechanisms by which exercise improves vascular function in diabetes are not fully understood. This study investigates whether there are (1) improvements in blood pressure and endothelium-dependent vasorelaxation (EDV) and (2) alterations in the relative contribution of endothelium-derived relaxing factors (EDRF) in modulating mesenteric arterial reactivity in male UC Davis type-2 diabetes mellitus (UCD-T2DM) rats, following an 8-week moderate-intensity exercise (MIE) intervention. EDV to acetylcholine (ACh) was measured before and after exposure to pharmacological inhibitors. Contractile responses to phenylephrine and myogenic tone were determined. The arterial expressions of endothelial nitric oxide (NO) synthase (eNOS), cyclooxygenase (COX), and calcium-activated potassium channel (K_Ca_) channels were also measured. T2DM significantly impaired EDV, increased contractile responses and myogenic tone. The impairment of EDV was accompanied by elevated NO and COX importance, whereas the contribution of prostanoid- and NO-independent (endothelium-derived hyperpolarization, EDH) relaxation was not apparent compared to controls. MIE 1) enhanced EDV, while it reduced contractile responses, myogenic tone and systolic blood pressure (SBP), and 2) caused a shift away from a reliance on COX toward a greater reliance on EDH in diabetic arteries. We provide the first evidence of the beneficial effects of MIE via the altered importance of EDRF in mesenteric arterial relaxation in male UCD-T2DM rats.

## 1. Introduction

The prevalence of type-2 diabetes (T2D) is rising globally due to obesity and physical inactivity on top of genetic predispositions, among other factors. Cardiovascular diseases (CVD) are the leading cause of mortality and morbidity in diabetic patients [1,2]. It is generally believed that T2D and CVD can be prevented and managed with healthy food choices and routine physical activity [3]. 

Increased CVD risk in diabetes is associated with vascular structural, mechanical, and functional alteration such as an increased wall to lumen ratio, stiffness and endothelial dysfunction. Vascular endothelial dysfunction is a hallmark of vascular disease; it is defined as reduced endothelium-dependent vasorelaxation (EDV) in response to vasodilators, such as acetylcholine (ACh). EDV is generally used as a reproducible parameter to investigate endothelial function under various pathological conditions. Impaired EDV has been reported in both type-1 and type-2 diabetes [4,5,6]. This could be due to the reduced production and/or release of endothelium-derived relaxing factors (EDRF). Some examples of these EDRF include nitric oxide (NO), prostacyclin (PGI_2_), and NO- and prostanoid-independent mediators. The contribution of various EDRF to EDV likely depends on the vascular beds. For example, NO is the major vasorelaxant in large conduit arteries. In small resistant arteries, NO- and prostanoid-independent or endothelium-dependent hyperpolarization (EDH) mediators are the major contributors [4,7]. In addition to EDRF, impaired EDV could be related to the increased production of endothelium-derived contractile factors (EDCF).

The pathology of cardiovascular dysfunction in T2D can be further elucidated with an appropriate animal model. This study utilizes a polygenic rodent model of T2D—the UC Davis type-2 diabetes mellitus (UCD-T2DM) rat model—in order to investigate vascular dysfunction in mesenteric arteries of this model. In particular, these rats exhibit polygenic adult-onset obesity, pancreatic beta-cell decompensation, preserved leptin signaling, and insulin resistance [8,9]. In our previous study, we demonstrated that in both large and small resistance arteries, NO and EDH-type contributions to EDV were altered in UCD-T2DM rats [6,10]. 

Exercise has beneficial effects on the cardiovascular system and is an important component of health related to the immune system [11,12]. Regular aerobic exercise is a well-known therapeutic intervention for endothelial dysfunction and CVD risk in T2D [13], yet the exact mechanisms by which exercise improves vascular function in diabetes remain poorly understood. Moreover, most studies have used large arteries to investigate the effects of exercise on vascular function and structure in diseased models; minimal attention has been given to the vital role that resistance arteries play in regulating blood pressure during exercise. Thus, the aim of our study was to determine whether the blood pressure and responses to endothelium-dependent vasodilators as well as vasoconstrictors in the mesenteric arteries of rats were affected after exercise. Specifically, we studied whether there were (1) improvements in blood pressure and EDV and (2) alterations in the relative contribution of EDRF in modulating mesenteric arterial reactivity in male UCD-T2DM rats, following an 8-week moderate-intensity exercise (MIE) intervention. We also investigated the effects of MIE on the responses to vasoconstrictor and the myogenic tone as well as on the arterial expression of endothelial nitric oxide (NO) synthase (eNOS), cyclooxygenase (COX), and intermediate conductance calcium-activated potassium channel (IK_Ca_) and small-conductance calcium-activated potassium channel (SK_Ca_) in male UCD-T2DM rats. 

We hypothesized that the improvement of arterial relaxation by MIE occurs via the alteration of the relative contributions of EDRF and/or EDCF in addition to the induction of changes in the arterial wall structure and myogenic tone in mesenteric arteries of this model of T2D.

This study demonstrates that 8 weeks of MIE improved systolic blood pressure (SBP) as well as vasorelaxation in the mesenteric arteries of male UCD-T2DM rats. Here, we provide the first report of the specific impact of MIE on the modulation of the relative importance of EDRF in vascular reactivity as well as myogenic tone and wall properties which could, in part, explain the improved EDV observed in the UCD-T2DM rat model.

## 2. Materials and Methods

### 2.1. Materials

All chemicals were purchased from either Fisher Scientific (Waltham, MA, USA) or Sigma-Aldrich (St. Louis, MO, USA), unless otherwise noted.

### 2.2. Experimental Animals

The UCD-T2DM rats were generated by selectively breeding obese Sprague Dawley (SD) rats with Zucker Diabetic Fatty (ZDF) lean rats at the animal facility in the Department of Nutrition at the University of California, Davis [8]. The model is well validated and characterized by >34 previous publications [9]. For this study, we selected 19–20 week-old male UCD-T2DM rats that had developed overt diabetes. Animals were considered diabetic when non-fasting blood glucose levels were higher than 300 mg/dL and exhibited the major sequelae of hyperglycemia: polyuria, polydipsia, and polyphagia. The diabetic animals used in the study were diabetic for 32 ± 2 days. Age-matched male SD rats were ordered from the Simonsen laboratory (CA). 

Only male rats were used for this study because of the relatively homogeneous metabolic phenotype of male animals compared with female animals in the UCD-T2DM and SD rat models. Furthermore, generating female UCD-T2DM rats at an age comparable to that of the male UCD-T2DM rats would have required the addition of a high fat/sugar diabetogenic diet. 

Both SD (control) and diabetic rats were randomly divided into either exercise-trained or sedentary groups. In total, four experimental groups were selected: (1) Control Sedentary (CS), (2) Control Exercise-Trained (CE), (3) Diabetic Sedentary (DS), and (4) Diabetic Exercise-trained (DE). 

All rats were maintained in a humidity- and temperature-controlled room with a 12 h light/dark cycle and ad libitum access to water and standard rodent chow (Mazuri rodent chow), except when fasting (described below). The euthanasia processes were completed in accordance with the AVMA Guidelines for the Euthanasia of Animals, 2013 edition, and the NIH Guidelines for the Care and Use of Laboratory Animals Eighth Edition. In addition, all animal protocols were approved by the Institutional Animal Care and Use Committee of the University of the Pacific. After the rats were sacrificed, the isolated mesenteric arterial function and wall characteristics were investigated using the wire myograph and pressure myograph technique.

### 2.3. Maximal Oxygen Consumption 

For acclimation, the exercise-trained groups were acclimatized with the treadmill for 5 days before the 8-week experimental MIE period by being placed on an unmoving treadmill for 5 min, after which running was motivated by a sporadic very low or minimum (0.4 mA) electrical shock at the rear of the apparatus and the treadmill was engaged to a walking speed of 15–18 m/min for 5 min. 

For the maximal oxygen consumption, after acclimation, the rats underwent MIE over the next 8 weeks. MIE is defined as the exercise intensity at which oxygen consumption reaches 50–65% of the maximum oxygen consumption (VO_2max_). The MIE protocol was selected based on prior published works [14,15]. Previous studies in rodents have used a treadmill speed ranging from 12–18 m/min with various inclinations for MIE [16,17,18]. In the current study, rats ran (5 days/week for 8 weeks) on a treadmill (Panlab Harvard App models, LE8706TS and LE8715TS) for 5 min at low speed, at 40–50% of the VO_2max_ (~10 m/min) for warm up, and then were subjected to an exercise session of 2 bouts of 25 min moderate-intensity running on a treadmill at 0° inclination, running at 55–60% of the VO_2max_ (~15–18 m/min) alternating with 5 min of active recovery (running at 40–50% of the VO_2max_) (~10 m/min). The sedentary group was placed on the stationary treadmill 3 times a week to provide a matched environment without MIE. 

### 2.4. Measurement of Metabolic Parameters in the Plasma

Blood glucose, triglycerides, plasma insulin, and glycated hemoglobin (HbA1c) were measured. We measured triglycerides and blood glucose levels in 12-h fasted rats via a drop of blood collected from the tail vein and specific test strips (Roche Farma, Barcelona, Spain). Triglycerides were measured with a point-of-care hand-held Accutrend Plus System device. Glucose was measured with a standard glucose test meter (OneTouch, LifeScan, Milpitas, CA, USA). After euthanizing the rats, additional blood samples were collected in tubes containing heparin and sodium citrate via intracardiac puncture. These tubes were centrifuged (at 4 °C, 10,000× *g* for 5 min), and then plasma was collected and aliquoted into new tubes, which were then stored at −80 °C for further analysis. Blood collected via intracardiac puncture was also used for HbA1c level analysis using the A1cNow kit (PTS diagnostics, Sunnyvale, CA, USA) in accordance with manufacturer instructions. Rats with HbA1c greater than 6.5% on two separate tests were considered diabetic. Plasma insulin levels were measured using ELISA kits compliant with the manufacturer’s protocol (Mercodia, Uppsala, Sweden).

### 2.5. Blood Pressure Measurement

Blood pressure in the unanesthetized rats was measured using the CODA^®^ noninvasive rat tail-cuff method (Kent Scientific, Torrington, CT, USA) following the manufacturer’s protocol and as previously described by others [19,20] and ourselves [21]. Briefly, rats were placed in restraints and heating chambers in order to be acclimatized to the dark, quiet and warm, quiet environment for about 30 min prior to the blood pressure measurement. SBP was measured between 10 a.m. and 3 p.m. prior to the start of exercise (0 week) and after 8 weeks of exercise. The rats were given a week to habituate to the procedure prior to the experiments. 

### 2.6. Measurement of Mesenteric Arterial Tension 

After isolating third-order mesenteric arteries from surrounding tissues, veins, and fat, the branches were cut precisely into 2 mm rings. Each ring was then mounted in an organ bath, between the two jaws of a wire myograph (model 610M, Danish Myo Technology (DMT), Hinnerup, Denmark) with the help of two tungsten wires (diameter 40 μm). The organ bath contained a Krebs solution of 119 mM NaCl, 25 mM NaHCO_3_, 1.2 mM KH_2_PO_4_, 1.2 mM MgSO_4_, 1.6 mM CaCl_2_, 4.7 mM KCl, 0.023 mM EDTA, and 6 mM glucose at 37 °C, bubbled with 95% O_2_–5% CO_2_. The variation in arterial isometric tension was monitored using a computer-based data acquisition system (Labchart version 7.3.8, Powerlab; ADInstruments, Colorado Springs, CO, USA). By using a built-in normalization module in the wire myograph (compliant with guidelines provided in the DMT manual and published reports), the arteries were normalized to a resting passive pressure of 13.3 kPa [22,23]. The micrometer was gradually increased until a pressure of approximately 13.3 kPa/100 mmHg was achieved. Importantly, the micrometer was not moved back after reaching 13.3 kPa, and thus our normalization factor was close to 1.0. The tissues were subsequently equilibrated for 30 min to establish a basal tone, and then an 80 mM KCl solution was used two times for a short period in order to stimulate the arterial segments. To evaluate endothelial viability, ACh (10 μM)-induced relaxation was recorded in phenylephrine (PE, 2 μM)-precontracted arteries. Following this, the drugs were rinsed out, the vessels were re-equilibrated for 30 min, and a cumulative concentration response curve (CRC) to PE (10^−8^ to 10^−5^ M) was performed. For the vasorelaxation studies, the arteries were precontracted with 2 μM of PE which induced approximately 80% of the contraction achieved by 10 μM of PE [22,23,24]. 

#### 2.6.1. Relaxation Responses to ACh

Increasing concentrations of ACh (10^−8^ to 10^−5^ M) were added to the PE (2 μM)-precontracted artery rings to obtain the CRC. The vascular relaxation response to ACh (10^−8^ to 10^−5^ M) in mesenteric arterial rings was then obtained after pretreatment with indomethacin (Indo, 10 μM, a blocker of the cyclooxygenase (COX)), followed by the addition of a Nω-nitro-L-arginine methyl ester (L-NAME, 200 μM, nitric oxide synthase (NOS) blocker) and then a combination of apamin (1 μM, small-conductance calcium-activated potassium channel (SK_Ca_) inhibitor) and 1-[(2-Chlorophenyl) diphenylmethyl]-1H-pyrazole (TRAM-34, 1 µM, intermediate conductance calcium-activated potassium channel (IK_Ca_) inhibitor) [24,25].

#### 2.6.2. Relaxation Responses to Sodium Nitroprusside (SNP)

The CRC to SNP (10^−9^ to 10^−5^ M) was obtained in mesenteric arteries precontracted with PE (2 μM) after 20 min preincubation with a combination of indomethacin (Indo, 10 µM), L-NAME (200 µM), apamin (1 µM) and TRAM-34 (1 µM). 

#### 2.6.3. Contractile Responses to PE 

The CRC to PE was obtained by the addition of increasing concentrations of PE (10^–8^ to 10^–5^ M) to the wire myograph chamber. The concentrations of drugs used to generate relaxation or contraction curves were based on our previous reports [4,6,25].

#### 2.6.4. Measurement of Myogenic Tone and Wall Thickness 

The third-order mesenteric arteries were isolated by clearing them from fat and other connective tissues (as described above) and mounted on the glass cannula of a DMT pressure myograph (model 114p). Both ends of the artery were secured on the cannulas by a thin nylon suture so that the artery and the cannulas together made a closed system. The mesenteric artery was then checked for any leak by pushing a small amount of the Krebs solution into the system. When the arteries maintained a bulbous formation, the system was considered a closed system. Care was taken not to introduce any bubbles into the system. The mesenteric artery was then pressurized to 160 mmHg and checked for any bends. Any bend observed in the mesenteric arterial segment was corrected by stretching the artery using the horizontal movement screw. The mesenteric arterial segment was allowed to rest at 60 mmHg for 20 min with a physiological saline solution (PSS) bubbled with 95% O_2_ -5% CO_2_ (at a pH of 7.35–7.40 in the bath) maintained at 37 °C, and then the viability of the arteries was determined by assessing ACh (10 µM)-mediated relaxation in the PE (2 µM)-precontracted arteries. The artery segment was washed and allowed to rest for 20 min before moving onto the pressurized experiments. 

Mesenteric arterial inner and outer diameter, wall thickness, and myogenic tone were then assessed in mesenteric arteries from the experimental groups over increasing intraluminal pressures in steps between 20 and 120 mmHg. Spontaneous tone was allowed to develop at each pressure until a stable diameter was achieved in a 5 min interval (active diameter, AD). Passive diameter (PD) at 60 mmHg in a Ca^2+^-free physiological salt solution was determined to calculate the myogenic tone at this pressure. The myogenic tone was expressed as a percent of the PD and calculated as (PD_60mmHg_ − AD_60mmHg_)/PD_60mmHg_ × 100 [24]. 

### 2.7. Western Blot Analysis

All tissue samples harvested after euthanizing the animals were flash frozen by liquid nitrogen and stored at −80 °C for later analysis. The mesenteric arteries were micronized using the gentleMACS tissue dissociator (Miltenyi Biotech, Bergisch, Germany), following the manufacturer’s protocol for protein extraction. A commercial RIPA buffer supplemented with a phosphatase and protease inhibitor cocktail (ThermoFisher Scientific, Waltham, MA, USA) was used to obtain the total protein extracted from the tissues. Briefly, for processing by the gentleMACS tissue dissociator, tissues were placed in M-tubes (Miltenyi Biotech, Bergisch, Germany) containing the RIPA buffer, phosphatase, and protease inhibitor cocktail. The protein extraction protocol was selected from the menu and after 1 min the blended tissue extract was centrifuged at 15,000× *g* for 15 min at 4 °C, and supernatants were collected. The total protein concentration of the extract was determined by a BCA gold assay (Thermo Fisher Scientific, Waltham, MA, USA). An amount of 20–30 μg of protein for each sample was loaded in sodium dodecyl sulfate poly acrylamide gels (SDS-PAGE) and subjected to gel electrophoresis. Protein was then transferred to 0.45 μm nitrocellulose membranes (Bio Rad Laboratories Inc., Hercules, CA, USA), blocked for 1 h at room temperature with 5% *w*/*v* BSA in 0.1% Tween 20-Tris-buffered saline (TBS), and incubated overnight at 4 °C with primary antibodies similarly to the previous method described by us [10,25]. Primary antibodies were obtained either from Cell Signaling Technology (Danvers, MA, USA) or Abcam (Cambridge, MA, USA). From Cell Signaling, the primary antibodies for endothelial nitric oxide synthase (eNOS) (#32027) and cyclooxygenase-1 (COX-1) (#4841S) were obtained. From Abcam (Cambridge, MA, USA), we obtained antibodies against cyclooxygenase-2 (COX-2) (#ab15191), IK_Ca_ (K_Ca_3.1) (#ab215990), and SK_Ca_ (K_Ca_2.3) (#ab220864). These primary antibodies were all diluted to 1:1000. The membranes were incubated with primary antibodies overnight. Then, the membranes were washed 4 times and incubated for 1 h at room temperature with the IRDye 680 Donkey anti-Rabbit IgG secondary antibody (diluted to 1:10,000, LI-COR, Lincoln, NE, USA). Finally, after the removal of the secondary antibody, the membranes were washed 4 times with TBS containing 0.1% Tween-20. The LICOR Odyssey imaging system was used to detect the bands. These bands were quantified via densitometry using LI-COR Image Studio Lite (version 5.2, Lincoln, NE, USA). Blots were incubated with GAPDH antibodies (#2118, Cell Signaling Technology, Danvers, MA, USA). In order to confirm uniformity in the protein loading, the blots were normalized to the GAPDH level and expressed as fold changes from the control group.

### 2.8. Statistical Analysis

Vasorelaxation in responses to ACh and SNP were expressed as the percentage of relaxation response from the maximum PE (2 μM) contraction at each concentration. Using a sigmoidal concentration response model with a variable slope via GraphPad Prism 8.0 (GraphPad Software, San Diego, CA, USA), the concentration that produced half of the maximum relaxation (EC_50_) was calculated. This was then expressed as sensitivity to the agonist, pD_2_ values (-logEC_50_). The maximum tension in response to the contractile agent (including PE) was expressed as Tension_max_. The maximum relaxation response to the agonist was expressed as R_max_. To compare the means between the different groups—for example: those of EC_50_, R_max_, blood glucose level, blood pressure—one-way ANOVA was used. When the one-way ANOVA returned a value of *p* of < 0.05, we determined which groups were different from each other using Tukey’s post hoc test. Student’s paired *t*-test was used for comparisons of blood pressure before MIE (pre-MIE at 0 week) and after MIE (post-MIE after 8 weeks) within a group. A two-way ANOVA with repeated measures followed by Tukey’s post hoc test was used to compare the CRCs between the different groups, with concentration being considered a repeated measure. A two-way ANOVA with repeated measures followed by Bonferroni’s post hoc analysis was used to compare the CRC before and after treatment with drugs within a group. A one-way ANOVA and Tukey’s post hoc test were used for a statistical analysis of protein expression. 

## 3. Results

### 3.1. Effects of MIE on Metabolic Parameters and Blood Pressure

MIE significantly reduced body weight in the diabetic group; however, this did not reach to the level observed in the control groups. The body weight was 383.3 ± 4.2 g in the CS group and 366.7 ± 5.1 g in the CE group, and 559.3 ± 30.4 g in the DS group and 469.3 ± 6.2 g in the DE group (Figure 1A). The intra-abdominal adipose tissue level (located around mesentery and omental) normalized to body weight was significantly higher in the DS group compared to the exercise-trained (CE and DE) groups (Figure 1B). Similarly, when circulating triglyceride levels were compared, the DS group showed significantly higher triglyceride levels (3.52 ± 0.3 mmol/L) compared to the CS, CE, and DE groups (1.35 ± 0.2 mmol/L, 1.36 ± 0.1 mmol/L, and 1.54 ± 0.2 mmol/L, respectively). However, MIE had no significant effect on blood glucose, HbA1c, and plasma insulin levels regardless of health status. Blood glucose and HbA1c levels were higher, and plasma insulin levels were lower in the diabetic groups than in the control groups (Figure 1D–F).

It has been shown that hypertension is an important cardiovascular risk factor [26]. We, therefore, measured blood pressure in UCD-T2DM rats prior to the start of exercise (0 week) and after 8 weeks of exercise. As shown in Table 1, systolic blood pressure (SBP), diastolic blood pressure (DBP) and mean arterial pressure (MAP) were significantly higher in the diabetic groups compared with those in the control animals prior to MIE (0 week). MIE significantly reduced SBP in the DE group compared to the DS group. Although the DBP and MAP were lower in the DE group than in the DS group, the differences were not statistically significant. 

When we compared the measurements of blood pressure before and after MIE within the respective groups, we observed a reduction in SBP in the CE group and reductions in SBP, DBP and MAP in the DE group (*p* < 0.05 vs. 0 week within the respective group; Student’s paired *t*-test). However, a reduction in MAP was also observed in the DS group after 8 weeks compared to 0 week. Therefore, other factors such as reduced stress due to acclimatization to the measurement of blood pressure could have contributed to the improvement in blood pressure observed in the animals. 

### 3.2. Effects of MIE on Relaxation Responses to ACh 

Both sensitivity (pD_2_) and maximal response (R_max_) to ACh were significantly lower in the mesenteric arteries from DS rats compared with those of the rats in the CS group. Interestingly, exercise enhanced the ACh response only in the mesenteric arteries of the diabetic group (DE group); the R_max_ to ACh in mesenteric arterial rings in the DE group (90.74 ± 2.4%) was markedly enhanced compared to that in the DS group (59.85 ± 2.2%) (Figure 2, Table 2). Similarly, mesenteric arteries from the DE group showed improved sensitivity to ACh (6.72 ± 0.1) compared to the respective sedentary group (5.80 ± 0.2).

### 3.3. Effects of MIE on the Relative Contribution of EDRF 

The relative contributions of COX metabolites, NO, and EDH mediators to the vasorelaxation induced by ACh were estimated by sequentially inhibiting COX, NOS, and a combination of small and intermediate-conductance calcium-activated potassium channel (SK_Ca_ and IK_Ca_) inhibitors as previously reported by us and others [25,27,28]. The inhibition of COX by Indo (10 µM) enhanced the ACh-induced R_max_ from 59.85 ± 2.2% to 98.61 ± 0.4% in arteries from the DS group only, suggesting the elevation of contractile COX metabolites in the DS group (Figure 3C, Table 3). The addition of L-NAME (200 µM) resulted in a reduction in ACh-induced vasorelaxation in arteries from both the CS and DS groups. However, the effect was more prominent (and almost completely blocked ACh relaxation) in the DS group (CS vs. DS group, Figure 3A,C). The difference in area under the curve (ΔAUC) between ACh CRCs before and after addition of inhibitor was calculated as described in our previous study [29]. Particularly, an intriguing observation was that MIE partially restored the loss of EDH-mediated relaxation in male UCD-T2DM rats (DS vs. DE group, Figure 3C,D). The R_max_ of ACh-induced vasorelaxation after Indo + L-NAME was 10.27 ± 3.7% and 38.58 ± 13.9% in the DS and DE groups, respectively. Additionally, the ΔAUC of Indo and L-NAME-resistant relaxation was significantly lower in the DS group (6.24 ± 1.83) compared to the DE group (48.38 ± 9.1) (Figure 3C,D, gray shaded area; Table 3, column of Indo + L-NAME + Apamin + TRAM-34). Furthermore, unlike in the DS group, the inhibition of COX with Indo did not affect ACh responses in mesenteric arteries of the DE group (DS vs. DE, Figure 3C,D).

### 3.4. Effects of MIE on COX, eNOS, and K_Ca_ Expression 

To investigate the possible mechanism by which the COX, NO, and EDH contribution to ACh-induced relaxation might have been altered in the diabetic male rats, the protein expression levels of COX-1 and COX-2, eNOS, and SK_Ca_ and IK_Ca_ in mesenteric arteries were measured using Western blotting. 

Arteries taken from the DS group exhibited a greater level of both COX isoforms than those taken from the exercise-trained groups (CE, and DE). As shown in Figure 4, the expression of both COX-1 and COX-2 was elevated in the mesenteric arteries of the DS group compared to those of the other experimental groups (although the COX-1 expression in DS arteries did not reach a significant level when compared with CS arteries).

There were no significant changes in the expression of eNOS in the mesenteric arteries of the controls (CS and CE groups) (Figure 5). However, according to the elevated importance of NO in ACh-induced relaxation in the mesenteric arteries of the diabetic groups (Figure 3C,D), the Western blot analysis revealed that the expression of eNOS was significantly elevated in the arteries of the diabetic animals (DS and DE groups) compared with the corresponding levels in the arteries of the controls (CS and CE groups) (4-folds and 3.25-folds, respectively) (Figure 5). 

The Western blot analysis also showed that there were no significant changes in the expression of IK_Ca_ in the mesenteric arteries of the experimental groups (Figure 6A). However, the level of SK_Ca_ expression was significantly decreased in the arteries of the DS group compared with those of the controls. On the other hand, the level of SK_Ca_ expression was significantly increased (1-fold) in the mesenteric arteries of the DE animals compared with those of the animals in the DS group (Figure 6B). 

### 3.5. Effects of MIE on the SNP-induced Relaxation 

To investigate whether impaired smooth muscle sensitivity to NO led to reduced mesenteric arterial relaxation in the DS rats, we examined the relaxation responses to SNP (NO donor) in mesenteric arteries precontracted with PE. There was no difference in the pD_2_ and R_max_ in response to SNP-induced relaxation regardless of the fact that MIE was performed and/or the health status in any of the experimental groups (Appendix A). 

### 3.6. Effects of MIE on the PE-induced Contraction

Next, we examined whether the impairment of ACh relaxation in the DS group was due to the altered contractile response of the arteries. The CRC to PE (10^−8^ to 10^−5^ M) in mesenteric arteries was, therefore, compared among experimental groups (Figure 7). The maximum contractile response to PE (Tension_max_) was higher in the arteries of the DS group compared to those in the controls and the DE group (Figure 7, Table 4). In contrast, the sensitivity to PE was not higher in the DS group. MIE significantly reduced the PE-induced maximal tension in the arteries of the diabetic rats (DE) to the same level as those of CS and CE rats.

### 3.7. Effects of MIE on the Myogenic Tone 

Microvasculature regulates blood flow to organs by virtue of the myogenic tone (constriction in response to intraluminal pressure) and therefore ensures smooth blood flow to organs despite changes in systemic hemodynamics. Mesenteric arterial myogenic tone was assessed over increasing intraluminal pressures (20–120 mmHg). The myogenic tone in DS arteries was significantly higher compared to that in the arteries in the other groups (Figure 8). MIE significantly reduced the myogenic tone in the DE group, although it did not reach the level observed in the control groups (the CS and CE groups). There was no difference in the myogenic tone (%) between CS and CE rats.

### 3.8. Effects of MIE on Wall Thickness

Increased intima–media thickness (IMT) could be an early marker of atherosclerosis [30]. Adverse mesenteric arterial remodeling in diabetic models has been reported [31,32]. In the current study, wall thickness was measured to investigate the effect of diabetes and MIE on mesenteric arterial remodeling. The mesenteric arteries taken from the DS group exhibited a significantly higher wall thickness compared to all other groups. However, the wall thickness of the DE group was comparable to that of the control groups. There was no difference in the mesenteric arterial wall thickness among the CS, CE, and DE groups (Figure 9).

## 4. Discussion

Here, we provide the first report of the impact of an 8-week MIE on the modulation of the relative importance of EDRF in vascular reactivity in mesenteric arteries of male UCD-T2DM rats. Specifically, our data indicate that MIE causes a shift from contractile COX to NO and EDH-type relaxation in diabetic rat arteries. This was in addition to the beneficial changes induced by MIE to the SBP, myogenic tone, and wall thickness in the mesenteric arteries of male UCD-T2DM rats.

Physical activity emerges as an effective strategy in the prevention and treatment of T2D [33]. A number of studies have demonstrated that exercise improves glucose homeostasis and insulin sensitivity [34,35,36]. According to the American College of Sports Medicine and American Diabetes Association, weekly 150 min of moderate to vigorous intensity exercise may help in losing weight [37,38]. However, 60 min of daily physical activity may be required when relying on exercise alone to lose weight [37]. Long-term adherence to exercise in individuals with T2D is problematic [39]. Here, we investigated the impact of short-term (8 weeks, 50 min/day, 5 days/week) of MIE on the potential improvement of vascular function in male UCD-T2DM rats. 

Impaired EDV as a measurable parameter of endothelial dysfunction is a hallmark of CVD in diabetes. Several studies, including ours, have reported the impairment of EDV in diabetes [4,6,40,41,42]. In accordance with our recent report [6] on the impairment of EDV in the mesenteric arteries of male UCD-T2DM, we showed that EDV was impaired in the mesenteric arteries of sedentary male UCD-T2DM rats; mesenteric arteries from DS rats exhibited a shift of ACh CRC to the right (Figure 2), along with a reduction in sensitivity to ACh.

The beneficial effects of exercise against vascular dysfunction in diabetes have been reported [43]. Sakamoto et al. reported that exercise training but not food restriction prevented endothelial dysfunction in type-2 diabetic Otsuka Long–Evans Tokushima fatty (OLETF) rat aorta, indicating exercise as the major physiological stimulus required for improving aortic function in diabetes [44]. In the current study, MIE improved the EDV response to ACh in the mesenteric arteries of the DE group compared to those of DS rats, as indicated by the increased R_max_ and sensitivity to ACh, without changing the glycemic status. Similarly, Moien-Afshari et al. reported that MIE enhanced the aortic function of db/db mice without changing hyperglycemic condition [41]. 

Insulin resistance plays an important role in the pathophysiology of T2D. Here, a comparable trend between vascular dysfunction and decreased insulin levels and subsequently increased glucose and HbA1C levels was observed in the DS group. Previous studies on different animal models of diabetes and insulin resistance demonstrated an impaired PI3K/AKT-dependent pathway in the vasculature of the tested animals, while ERK1/2 signaling was maintained or enhanced [45,46,47]. Accordingly, we recently reported a reduction in the p-AKT level in the mesenteric arteries of male UCD-T2DM rats [6]. In the current study, MIE did not alter the plasma insulin, blood glucose, and HbA1c levels in either the control or diabetic groups. In agreement with our observation, Knudsen et al. reported that the level of insulin was not changed after a single bout of MIE in T2D patients [48]. On the other hand, our data revealed that 8 weeks of MIE reduced body weight, abdominal adiposity, the triglyceride level as well as SBP in the DE rats compared to their sedentary counterparts (DS). Previous studies also reported that changes in triglyceride levels were observed in both ZDF rats and patients of coronary heart diseases after 8 weeks of aerobic exercise training [49,50]. 

Hypertriglyceridemia is shown to be an independent risk factor for CVD, and it contributes to the development of hypertension [51]. Elevated serum triglyceride increases endothelial cell permeability followed by increasing TNF-α secretion, adhesion molecule expression and oxidative stress. Thus, the proatherogenic and proinflammatory properties of triglyceride molecules lead to endothelial dysfunction [52,53]. A number of studies in animal models as well as in humans reported an inverse relationship between circulating triglyceride levels and endothelial functions [51,54,55,56]. In similar lines, we previously reported an impairment of EDV in both aorta and mesenteric arteries of Sprague–Dawley rats supplemented with a high-fructose, but not high-glucose, diet [21,29]. In the same reports, we demonstrated that the level of triglyceride concentration was significantly increased only in blood taken from fructose-supplemented rats, indicating that elevated triglyceride levels could be involved in endothelial dysfunction. This theory is further supported in the present study by the observation of the improvement of both plasma triglyceride levels and EDV by MIE, suggesting that the improvement of endothelial function in the DE group could be explained, in part, by the decreased triglyceride level in this group. 

Endothelial function is regulated differently in large vessels compared to small vessels. Although NO is generally considered the principal mediator of EDV, it has become clear that EDH is an important contributor to EDV, especially in small arteries such as mesenteric arteries. Tomioka et al. reported that the EDH contribution to EDV is inversely proportional to the vessel diameter [57]. 

We and other investigators have shown that both NO-dependent and NO-independent mechanisms are involved in vasorelaxation in rat mesenteric arteries [4,58]. Nevertheless, the specific roles of NO and NO-independent pathways (COX and EDH) on the beneficial impact of exercise training in pathophysiological conditions such as diabetes are less clear. 

Here, incubation of the DS arteries with indomethacin increased the maximum relaxation response to ACh, suggesting that the contribution of COX (likely its contractile metabolites) in modulating the vascular reactivity of mesenteric arteries was enhanced in the DS group (Figure 3C). In the presence of Indo and L-NAME, the decrease in vasorelaxation is considered to represent the role of NO, and the remaining vasorelaxation response to ACh is referred to as the L-NAME/Indo-insensitive component, or EDH-type relaxation [28,59,60]. Recent studies demonstrated that K_Ca_ channels are important for EDH-type vasorelaxation in rat mesenteric arteries. Although the K_Ca_ current was not measured in the current study design, our data on the sensitivity of the remaining L-NAME/Indo-insensitive component to the combination of apamin and TRAM-34 suggest that SK_Ca_ and IK_Ca_ channels were critical for EDH-type vasorelaxation in the mesenteric arteries of the experimental groups [61].

We previously demonstrated that EDH-type relaxation was diminished, whereas the dependency on NO-mediated relaxation was enhanced in diabetic mesenteric arteries [4,6]. Similarly, in the current study, we observed a loss of EDH-type relaxation with an increased dependency on NO-mediated relaxation in the mesenteric arteries of the DS group (Figure 3C). Specifically, the addition of L-NAME in all of the experimental groups led to a further decrease in EDV (Figure 3A–D). However, there was a more prominent blunting of ACh-mediated vasorelaxation due to the added effect of L-NAME in the arteries of diabetic rats. These data suggest that, in the diabetic groups, regardless of their physical activity status, the NO contribution to the mesenteric arterial relaxation responses was elevated (Figure 3C,D; Table 3). Notably, compared with the control groups, L-NAME completely blocked the remaining relaxation responses to ACh in the arteries from the DS group (Figure 3A–C).

The increased NO contribution may be explained by the elevated eNOS expression (Figure 5) observed in arteries from the DS and DE groups. Our finding is in agreement with the findings of previous reports which described increased eNOS expression in the heart and aorta of a type-2 diabetic Goto- Kakizaki rat model [62,63]. Despite the higher relative importance of NO and elevated eNOS expression, vascular relaxation was significantly impaired in the DS group, possibly due to loss of EDH (a major mediator of vasorelaxation in small arteries). Furthermore, any increase in NO has the potential for free radical-mediated damage, particularly under conditions of oxidative stress in diabetes, in which peroxynitrite is formed more easily [4]. It is also important to note that the augmented L-NAME (a nonselective NOS inhibitor) responses in the arteries from the diabetic rats (Figure 3C,D) may have also involved uncoupled eNOS, a major source of vascular superoxide in diabetes [64]. 

The elevated NO contribution in the diabetic mesenteric arteries could have been a compensatory mechanism for the loss of EDH-type relaxation. Previous studies suggested that an inverse feedback relationship exists between NO and EDH. NO reduced the EDH release in rabbit carotid and porcine coronary arteries [65,66]. Moreover, an elevated EDH-mediated response was reported to compensate for decreased NO-mediated relaxation in arteries from diabetic rats [67,68]. Kilpatrick and Cocks reported that in the absence of basal NO production, the L-NAME-insensitive (EDH) component of vasorelaxation was upregulated [69]. Here, our data suggest that the inhibition of NOS by L-NAME does not lead to an increase in EDH importance in EDV of DS arteries (Figure 3C). On the other hand, following NOS inhibition, we showed a complete loss of the EDH contribution to ACh relaxation, possibly due to the significant decrease in SK_Ca_ expression in the DS arteries (Figure 3C and Figure 6). Previously, we and others have also reported an attenuation in EDH response in mesenteric arteries from type-2 diabetic animals [6,70]. Specially, in pathological conditions such as hypertension, atherosclerosis, and diabetes, altered EDH-mediated relaxation was reported [71]. 

Here, we have demonstrated a shift of NO and EDH in arteries of normoglycemic control groups to COX and NO in arteries from DS rats. Unlike in the DS group, the inhibition of COX with Indo had no effect on the ACh response of the DE group, while the inhibition of NOS by L-NAME reduced (but did not completely block) ACh relaxation. Intriguingly, these data suggest that MIE caused a loss of the COX contribution while partially restoring EDH-mediated relaxation (possibly via enhanced SK_Ca_ expression) in the mesenteric arteries of the DE rats (DS vs. DE, Figure 3C vs. Figure 3D). Along similar lines, Minami et al. reported that 9 weeks of exercise training improved EDH contribution in the mesenteric arteries of type-2 diabetic OLETF rats [72]. Although both studies disclose an elevation in EDH in exercise-trained diabetic vasculature, here we also observed a shift away from COX to EDH-mediated relaxation in arteries from the DE group. The OLEFT rats were found to develop a satiety disorder due to the lack of CCK-1 receptors and the upregulation of neuropeptide Y (NPY) [73]. In humans, neither diabetes nor metabolic syndrome are monogenetic disorders. Here, we utilized the UCD-T2DM rats which possess the polygenic adult onset of diabetes with preserved leptin signaling.

Overall, in this study, we were not able to clearly delineate the reasons for the shift in the roles of COX and EDH in the vasorelaxation of the mesenteric arteries. Notably, COX and EDH do not only function in the regulation of vasomotor tone, but also have inflammatory and anti-inflammatory functions, respectively [74,75,76]. Thus, one might speculate that the impairment of mesenteric arterial relaxation for the DS rats may be also partly attributed to the enhanced inflammatory metabolite of COX and reduced anti-inflammatory effects of EDH. 

While COX-1 is constitutively expressed in the majority of tissues, COX-2 is mainly induced and expressed under inflammatory conditions. In spite of some reports of involvement of both isoforms of COX in endothelium-dependent contraction, a number of reports suggest that under pathological conditions such as obesity and hypertension, COX-2 could be considered a more prominent source of EDCF [77,78,79,80]. It was reported that COX-2 contributed to cardiac oxidative stress and coronary vasoconstriction in obese Zucker rats [81]. In hypertension, an overexpression of COX-2 and NADPH oxidase (NOX)-mediated release of reactive oxygen species (ROS) led to enhanced endothelium-dependent contractions [7]. Here, we did not use the selective inhibitor of COX or measure the COX-derived prostanoid; nevertheless, we observed an increase in both COX isoforms’ protein expression in DS arteries (although the difference in COX-1 expression did not reach a significant level). These data suggest that the enhanced COX-derived contractile metabolites could have been in part a mediator of the impaired vasorelaxation in the DS group. Additionally, consistent with our working hypothesis are the data demonstrating that the mesenteric arteries from DS group exhibited higher maximal contractile responses to PE, compared with other groups (Figure 7), and that MIE reduced the PE-induced maximal tension to the same level as that in the control animals (DE vs. CS group or DE vs. DS group, Figure 7). Accordingly, in the current study, the expression levels of both COX-1 and COX-2 were significantly decreased in the mesenteric arteries of the DE group compared with the DS group. 

Endothelial cells may be indirectly stimulated by PE to release NO via a signal that may be transmitted by mechanical stress [82] or through myoendothelial gap junctions [83,84]. During smooth muscle contraction, the decreased release of NO from the endothelium may have therefore led to the elevated contractile responses to PE in the mesenteric arteries of the DS group [4,10]. Here, we did not measure NO or contractile prostanoids. Nevertheless, similarly to the increased PE responses in the DS group, we demonstrated that myogenic tone in the mesenteric arteries of the DS group was enhanced, and MIE significantly decreased the myogenic tone in DE rats (although it did not reach the level observed in the control groups, Figure 8). This is consistent with earlier reports on enhanced myogenic tone in mesenteric, cerebral, and skeletal muscle arteries in T2D rodent models [85,86,87]. Previous investigators reported that an increase in the levels of one or more vasoconstricting prostanoids could lead to an increase in myogenic tone in diabetic animals, and they suggested that COX-2 could be a major factor contributing to increased arterial tone in those models [88]. Furthermore, Leo et al. recently reported that an upregulation of TMEM16A, a calcium-activated chloride channel, in resistance-sized arterial smooth muscle cells of T2D mice could lead to greater pressure-induced vasoconstriction (myogenic response) than in the arteries of nondiabetic control mice [89]. Thus, the augmented myogenic response observed in the mesenteric arteries of the UCD-T2DM rats may have involved the upregulation of TMEM16A in the smooth muscle cells of this model. Clearly, further studies are required to elucidate the role of NO, vasoconstricting prostanoids, specific COX isoforms as well as the TMEM16A channels in mesenteric arterial function in UCD-T2DM rats following exercise.

Diabetes-associated vascular diseases manifest endothelial dysfunction followed by structural and functional changes in the resistance of small arteries, leading to increased peripheral resistance, which is associated with hypertension [90,91,92]. IMT is considered a surrogate marker of atherosclerosis in pathogenic conditions [93]. In mesenteric arteries from db/db mice, increased outward remodeling was characterized by an enhanced cross-sectional area, thus suggesting an increase in wall thickness in arteries from diabetic rats [32]. Briones et al. reported that increased wall thickness, together with increased collagen deposition and a change in the elastic lamina, was associated with increased stiffness in mesenteric arteries from aged rats [94]. In the current study, we demonstrated that diabetes significantly increased the mesenteric arterial wall thickness in the DS group. It has been also reported that endothelial dysfunction in T2D is accompanied by the increased production of vasoactive compounds such as ET-1, angiotensin II, oxidative stress, and growth factors, ultimately leading to the elevation in vasoconstriction, myogenic tone and vascular wall thickening [95,96]. Here, we did not measure any of those contracting factors; however, the observed elevation in vasoconstriction, myogenic tone, and vascular wall thickening may have contributed to the increased blood pressure that was detected in the DS rats. On the other hand, it is possible that the reduction in blood pressure and triglyceride level following MIE may have contributed to the improvements in vascular and endothelial function in the DE group compared with the DS animals. In agreement with our observation of reduced IMT and SBP by MIE, aerobic exercise reduced mesenteric arterial wall thickness (media/lumen ratio) and blood pressure in spontaneously hypertensive rat (SHR) model [97].

Finally, another remaining factor that may have contributed to improved mesenteric arterial relaxation in DE group is the increased sensitivity of mesenteric arteries to NO. Here, however, we did not observe any differences in SNP-induced relaxation among the mesenteric arteries taken from the four experimental groups. This may suggest that smooth muscle sensitivity to NO was unaffected by diabetes or exercise. Notably, in the current study SNP responses were measured in the presence of Indo + L-NAME + Apamin + TRAM-34 to rule out the possible inhibitory effects of basal NO and other EDRF on SNP response [25,98,99,100].

## 5. Conclusions

This study represents the first report showing that MIE improves mesenteric arterial function in part by (1) reducing plasma triglyceride levels, (2) inducing a shift away from contractile COX to NO- and EDH-type relaxation, (3) reducing vasocontractile response and myogenic tone, and (4) preventing adverse arterial remodeling by reducing wall thickness, leading to decreased SBP in male UCD-T2DM rats. Clearly, additional studies are required to further elucidate the relationship between diabetes and the beneficial impact of exercise on vascular functions and wall properties. Future studies on female animals will also permit an assessment of the potential sex-specific impact of exercise on vascular reactivity in UCD-T2DM rats. In addition, how these findings relate to the impact of MIE on other organs such as skeletal muscle and adipose tissue in UCD-T2DM rats should be investigated. 

## Figures and Tables

**Figure 1 biomedicines-11-01129-f001:**
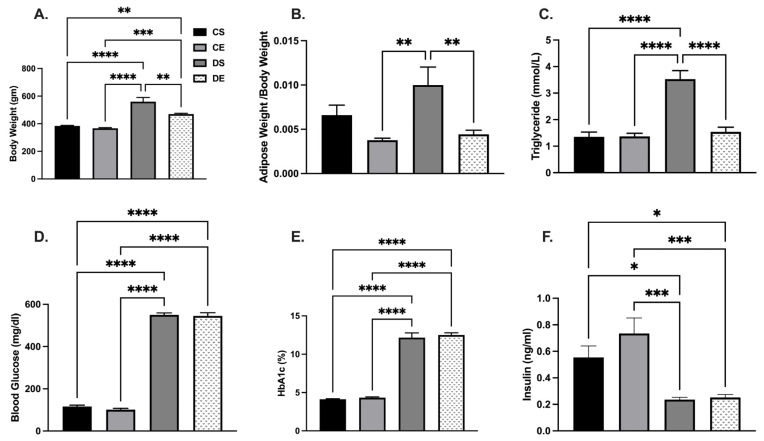
Effects of MIE on metabolic parameters of sedentary and exercise-trained control (CS and CE) and diabetic (DS and DE) rats. (**A**) Body weight; (**B**) adipose tissue to body weight ratio; (**C**) triglyceride level; (**D**) blood glucose level; (**E**) HbA1c (%); (**F**) insulin level. Values are represented as mean ± SEM. Each bar represents the values obtained from n = 5–7 animals per group. Capped lines indicate significant differences between two groups (* *p* < 0.05, ** *p* < 0.01, *** *p* < 0.001, and **** *p* < 0.0001), analyzed by one-way ANOVA followed by Tukey’s post hoc test.

**Figure 2 biomedicines-11-01129-f002:**
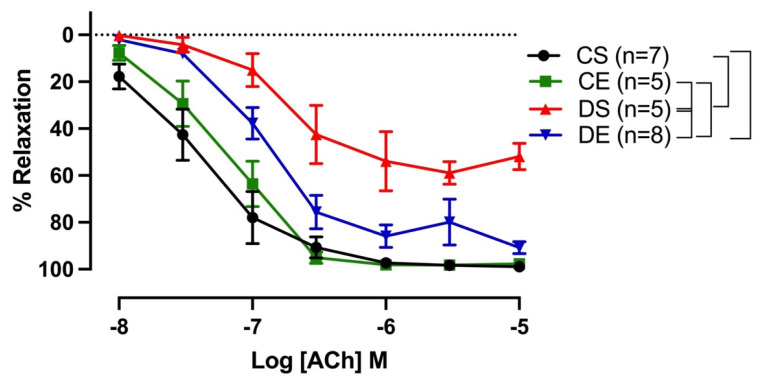
Relaxation responses to cumulative concentrations of acetylcholine (ACh, 10^−8^ to 10^−5^ M) in intact mesenteric arterial rings precontracted with phenylephrine (PE, 2 µM) from sedentary and exercise-trained control (CS and CE) and diabetic (DS and DE) rats. Data are expressed as mean ± SEM; n = 5–8 animals per group. Brackets indicate significant differences between groups (*p* < 0.05), analyzed using two-way ANOVA with repeated measures followed by Tukey’s post hoc test.

**Figure 3 biomedicines-11-01129-f003:**
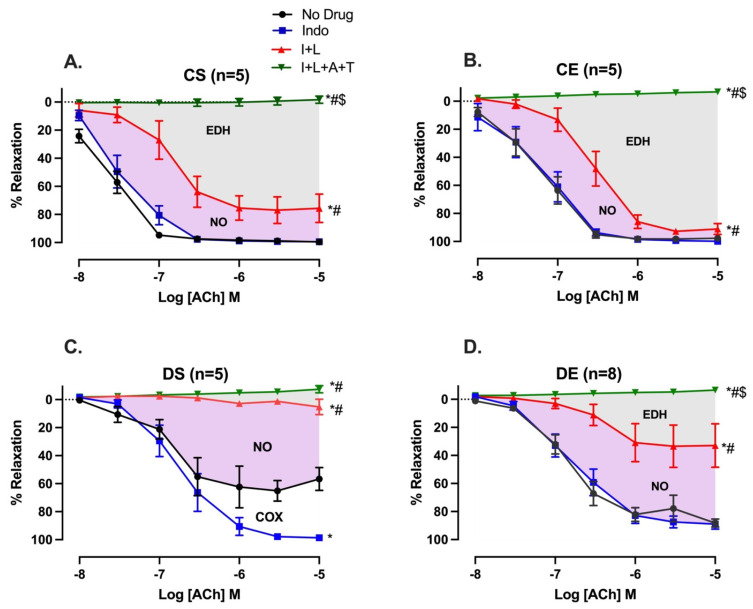
Effects of inhibiting cyclooxygenase (COX), nitric oxide synthase (NOS), small-conductance and intermediate-conductance calcium-activated potassium channels (SK_Ca_ and IK_Ca_) on acetylcholine (ACh)-induced vasorelaxation in mesenteric arteries taken from sedentary and exercise-trained control (CS and CE) and diabetic (DS and DE) rats; (**A**) CS, (**B**) CE, (**C**) DS, and (**D**) DE group. CRC to ACh (10^−8^ to 10^−5^ M) was generated in the absence of an inhibitor (no drug) or in the presence of Indo, Indo + L-NAME (I + L), and Indo + L-NAME + Apamin + TRAM-34 (I + L + A + T) in mesenteric arterial rings. Indo (indomethacin; 10 µM), L-NAME (N^ω^-nitro-L-arginine methyl ester; 200 µM), Apamin (1 µM), TRAM-34 [1-[(2-Chlorophenyl) diphenylmethyl]-1H-pyrazole; 1 µM]. * (*p* < 0.05) vs. no drug; # (*p* < 0.05) vs. Indo; $ (*p* < 0.05) vs. Indo + L-NAME, analyzed using two-way ANOVA with repeated measures followed by Bonferroni post hoc test.

**Figure 4 biomedicines-11-01129-f004:**
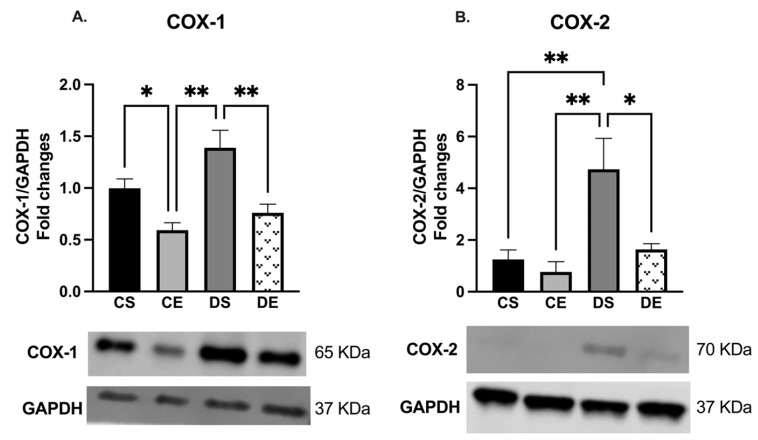
Western blot analysis of (**A**) COX-1 and (**B**) COX-2 expression in mesenteric arteries of sedentary and exercise-trained control (CS and CE) and diabetic (DS and DE) rats. Protein levels were quantified by densitometry and normalized to the corresponding GAPDH. Values are represented as mean ± SEM. Each bar represents the values obtained from n = 3–4 animals per group. Representative bands of target and housekeeping proteins (GAPDH) were shown from the same membrane. Capped lines indicate significant differences between two groups (* *p* < 0.05, and ** *p* < 0.01), analyzed using one-way ANOVA followed by Tukey’s post hoc test. COX-1, cyclooxygenase-1; COX-2, cyclooxygenase-2.

**Figure 5 biomedicines-11-01129-f005:**
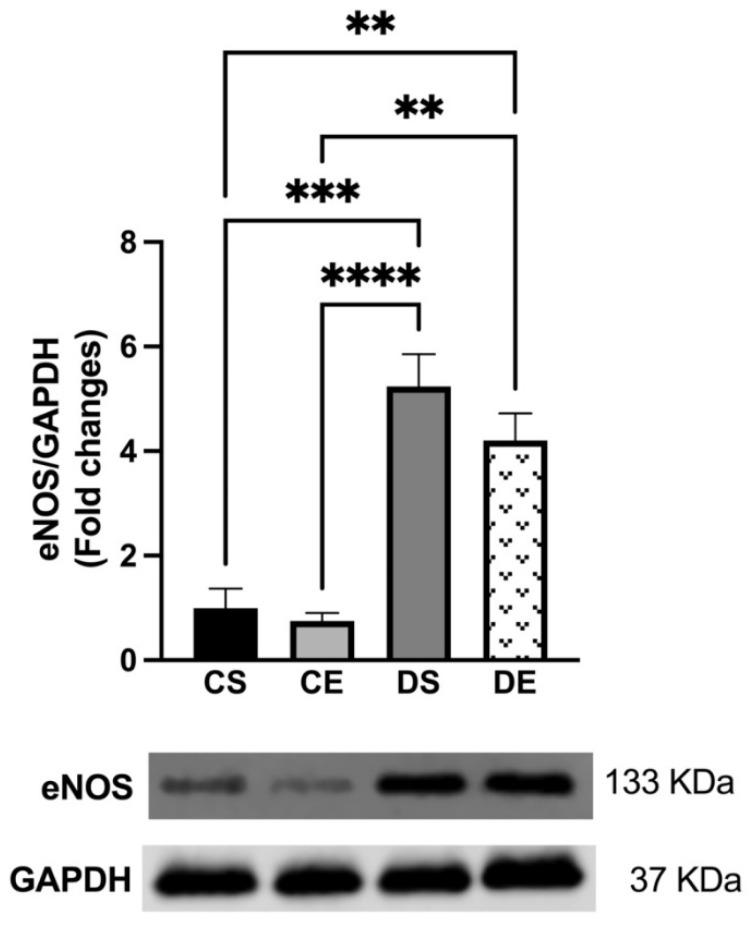
Western blot analysis of eNOS expression in mesenteric arteries of sedentary and exercise-trained control (CS and CE) and diabetic (DS and DE) rats. Protein levels were quantified by densitometry and normalized to the corresponding GAPDH. Values are represented as mean ± SEM. Each bar represents the values obtained from n = 4–5 animals per group. Representative bands of target and housekeeping proteins (GAPDH) were shown from the same membrane. Capped lines indicate significant differences between two groups (** *p* < 0.01, *** *p* < 0.001, and **** *p* < 0.0001), analyzed using one-way ANOVA followed by Tukey’s post hoc test. eNOS, endothelial nitric oxide synthase.

**Figure 6 biomedicines-11-01129-f006:**
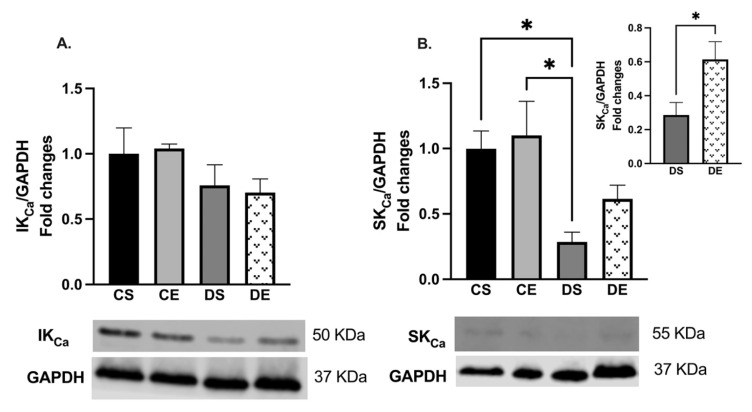
Western blot analysis of (**A**) IK_Ca_ (K_Ca_3.1) and (**B**) SK_Ca_ (K_Ca_2.3) expressions in mesenteric arteries of sedentary and exercise-trained control (CS and CE) and diabetic (DS and DE) rats. Protein levels were quantified by densitometry and normalized to the corresponding GAPDH. Values are represented as mean ± SEM. Each bar represents the values obtained from n = 4–5 animals per group. Representative bands of target and housekeeping proteins (GAPDH) were shown from the same membrane. Capped lines indicate significant differences between two groups (* *p* < 0.05), analyzed by one-way ANOVA followed by Tukey’s post hoc test. The top right section of B shows a comparison of only diabetic groups (DS vs. DE) which was analyzed by Student’s unpaired *t*-test. IK_Ca_, intermediate conductance calcium activated potassium channel; SK_Ca_, small conductance calcium activated potassium channel.

**Figure 7 biomedicines-11-01129-f007:**
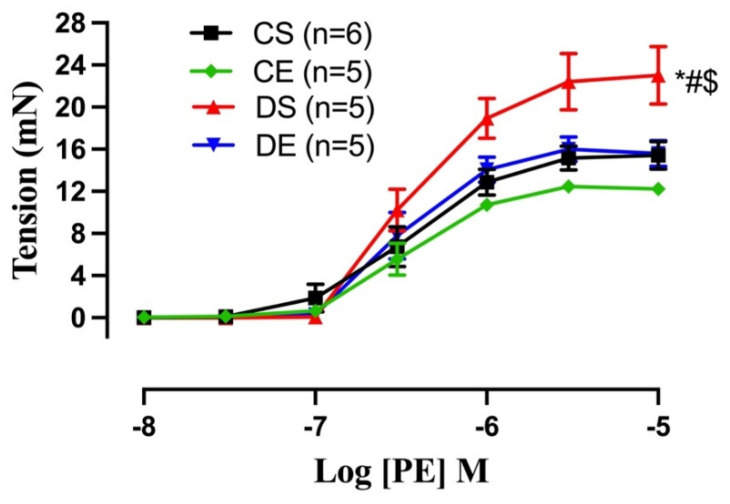
Contractile responses to cumulative concentrations of phenylephrine (PE, 10^−8^ to 10^−5^ M) in intact mesenteric arterial rings from sedentary and exercise-trained control (CS and CE) and diabetic (DS and DE) rats. Data are expressed as mean ± SEM; n = 5–6 animals per group. # *p* < 0.05 (vs. CS), * *p* < 0.05 (vs. CE), and $ *p* < 0.05 (vs. DE), analyzed using two-way ANOVA with repeated measures followed by Tukey’s post hoc test.

**Figure 8 biomedicines-11-01129-f008:**
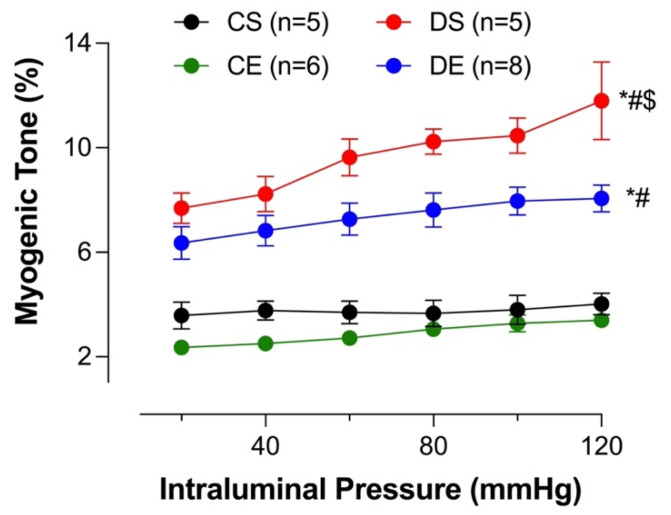
Myogenic tone (%) at different intraluminal pressures in intact mesenteric arterial rings from sedentary and exercise-trained control (CS and CE) and diabetic (DS and DE) rats. Data are expressed as mean ± SEM; n = 5–8 animals per group. # *p* < 0.05 (vs. CS), * *p* < 0.05 (vs. CE), and $ *p* < 0.05 (vs. DE), analyzed using two-way ANOVA with repeated measures followed by Tukey’s post hoc test.

**Figure 9 biomedicines-11-01129-f009:**
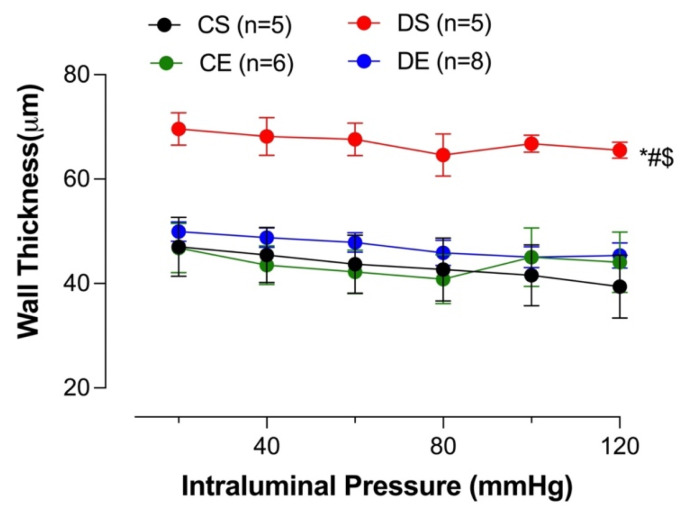
Wall thickness at different intraluminal pressures in intact mesenteric arterial rings from sedentary and exercise-trained control (CS and CE) and diabetic (DS and DE) rats. Data are expressed as mean ± SEM; n = 5–8 animals per group. # *p* < 0.05 (vs. CS), * *p* < 0.05 (vs. CE), and $ *p* < 0.05 (vs. DE), analyzed using two-way ANOVA with repeated measures followed by Tukey’s post hoc test.

**Table 1 biomedicines-11-01129-t001:** Blood pressure before and after 8 weeks in sedentary and exercise-trained control (CS and CE) and diabetic (DS and DE) rats.

Time	mmHg	CS	CE	DS	DE
0 week	SBP	114 ± 3	119 ± 6	155 ± 8 *#	153 ± 5 *#
	DBP	80 ± 2	86 ± 3	117 ± 8 *#	112 ± 6 *#
	MAP	95 ± 3	99 ± 4	129 ± 8 *#	125 ± 6 *#
8 weeks	SBP	114 ± 3	98 ± 4 ^a^	138 ± 5 *#$	115 ± 5 ^a^
	DBP	76 ± 2	84 ± 10	93 ± 3	79 ± 4 ^b^
	MAP	91 ± 3	85 ± 8	104 ± 7 ^c^	91 ± 4 ^c^

Data are expressed as mean ± SEM; n = 5–8 rats per group. Data were analyzed between groups using one-way ANOVA followed by Tukey’s post hoc test; # *p* < 0.05 (vs. CS), * *p* < 0.05 (vs. CE), and $ *p* < 0.05 (vs. DE). Data were compared within a group (0 week vs. 8 weeks) by Student’s paired *t*-test. ^a^
*p* < 0.05 (vs. 0 week SBP, respective group), ^b^
*p* < 0.05 (vs. 0 week DBP, respective group), ^c^
*p* < 0.05 (vs. 0 week MAP, respective group). SBP, systolic blood pressure; DBP, diastolic blood pressure; and MAP, mean arterial pressure.

**Table 2 biomedicines-11-01129-t002:** Sensitivity (pD_2_:-logEC_50_) and maximum response (R_max_) to acetylcholine (ACh) in mesenteric arteries from sedentary and exercise-trained control (CS and CE) and diabetic (DS and DE) rats.

ACh	n	pD_2_ (-logEC_50_)	R_max_ (%)
CS	7	7.40 ± 0.2	98.91 ± 0.6
CE	5	7.22 ± 0.1	98.60 ± 1.0
DS	5	5.80 ± 0.2 *#$	59.85 ± 2.2 *#$
DE	8	6.72 ± 0.1 #	90.74 ± 2.4 *#

Data are expressed as mean ± SEM; n = 5–8 rats per group. # *p* < 0.05 (vs. CS), * *p* < 0.05 (vs. CE), and $ *p* < 0.05 (vs. DE), analyzed using one-way ANOVA followed by Tukey’s post hoc test.

**Table 3 biomedicines-11-01129-t003:** Comparison of sensitivity (pD_2_: -logEC_50_), maximum response (R_max_), and area under the curve (AUC) to acetylcholine (ACh) in mesenteric arteries from sedentary and exercise-trained control (CS and CE) and diabetic (DS and DE) rats.

Groups	No Drug			Indo			Indo + L-NAME			Indo + L-NAME + Apamin + TRAM-34		
	pD_2_	R_max,_ (%)	ΔAUC	pD_2_	R_max_ (%)	ΔAUC	pD_2_	R_max_ (%)	ΔAUC	pD_2_	R_max_ (%)	ΔAUC
CS	7.63 ± 0.1	99.57 ± 0.1	ND	7.49 ± 0.1	99.41 ± 0.2	13.80 ± 5.71	ND	80.54 ± 7.3	93.90 ± 9.62	ND	−1.71 ± 2.5 ^abc^	147.3 ± 8.34
CE	7.22 ± 0.12	98.60 ± 0.9	ND	7.26 ± 0.2	99.83 ± 0.2	1.0 ± 7.66	ND	93.88 ± 1.9	75.20 ± 8.13	ND	−6.67 ± 1.4 ^abc^	144.7 ± 5.63
DS	5.80 ± 0.2 *#$	59.85 ± 2.2 *#$	ND	6.70 ± 0.2 #^a^	98.61 ± 0.4 ^a^*#$	47.20 ± 4.77 *#$	ND	10.27 ± 3.7 *#^ab^	115.9 ± 3.45 *	ND	−7.36 ± 2.5 ^abc^	6.24 ± 1.83 *#$
DE	6.72 ± 0.1 #	90.74 ± 2.4 *#	ND	6.61 ± 0.1 *#	88.92 ± 3.4	0.5 ± 7.49	ND	38.58 ± 13.9 *#^ab^	107.7 ± 10.1	ND	−6.5 ± 1.4 ^abc^	48.38 ± 9.1 *#

Comparison of sensitivity (pD_2_), maximum response (R_max_) and ΔAUC to acetylcholine (ACh) in the absence (no drug condition) or in the presence of Indo, Indo + L-NAME, and Indo + L-NAME + Apamin + TRAM-34 in mesenteric arterial rings from sedentary and exercise- trained control (CS and CE) and diabetic (DS and DE) rats. ΔAUC values were measured by measuring the area between the two curves. ΔAUC of Indo indicates COX contribution, ΔAUC of Indo + L-NAME indicates NO contribution and, ΔAUC of Indo + L-NAME + Apamin + TRAM-34 indicates EDH-type contribution in ACh-induced vasorelaxation of mesenteric arteries. Data are expressed as mean ± SEM; n = 5–8 rats per group. Analysis between group: # *p* < 0.05 (vs. CS), * *p* < 0.05 (vs. CE), and $ *p* < 0.05 (vs. DE), analyzed using one-way ANOVA followed by Tukey’s post hoc test. Analysis within group: ^a^
*p* < 0.05 vs. no drug control within each group, ^b^
*p* < 0.05 vs. Indo within each group, ^c^
*p* < 0.05 vs. Indo + L-NAME within each group (paired Student’s *t*-test). ND; not determined.

**Table 4 biomedicines-11-01129-t004:** Sensitivity (pD_2_: -logEC_50_) and maximum tension (Tension_max_) to phenylephrine (PE) in mesenteric arteries from sedentary and exercise-trained control (CS and CE) and diabetic (DS and DE) rats.

PE	n	pD_2_(-logEC_50_)	Tension_max_
CS	6	6.49 ± 0.16	15.45 ± 1.24
CE	5	6.46 ± 0.12	12.23 ± 0.54
DS	5	6.46 ± 0.18	23.03 ± 2.74 *#$
DE	5	6.41 ± 0.11	16.20 ± 1.18

Data are expressed as mean ± SEM; n = 5–6 rats per group. # *p* < 0.05 (vs. CS), * *p* < 0.05 (vs. CE), $ *p* < 0.05 (vs. DE), analyzed using one-way ANOVA followed by Tukey’s post hoc test.

## Data Availability

The datasets generated during the current study are available from the corresponding author upon reasonable request.

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
