# Peer review of "Moderate-Intensity Exercise Improves Mesenteric Arterial Function in Male UC Davis Type-2 Diabetes Mellitus (UCD-T2DM) Rats: A Shift in the Relative Importance of Endothelium-Derived Relaxing Factors (EDRF)"

_biomedicines, 2023, doi:10.3390/biomedicines11041129_

Round 1

Reviewer 1 Report

How important is the insulin resistance in the reactivity of the mesenteric arteries. Please add to discussions.

Did you notice any differences in vascular reactivity between male and female?

What was the rationale to check relaxation to SNP? Did you expect impairment of the muscular layer?

What about the presence of atherosclerotic plaques in diabetic rats?

Reviewer 2 Report

In this paper, Razan et al. showed that moderate intensity exercise improves mesenteric arterial function in male UC Davis Type-2 diabetes mellitus rats mainly both by suppression of COX-mediated EDCF and by augmentation of the production of NO and EDH. This is a well-organized and well-written paper about an important topic. The authors’ findings are novel and interesting; however, this reviewer has several concerns that should be adequately addressed.

 Major comments

1.      The authors’ results suggest that the improvement of the vascular function after moderate intensity exercise seems independent of the blood glucose levels and the insulin resistance. Considering the fact that some models of obese rats are hypertensive, I think that the beneficial effects of the moderate intensity exercise could be due to the blood pressure lowering effect of the exercise. Could the authors provide the blood pressure data before and after the intervention in each group?    

2.      It is interesting that triglyceride levels are increased in this model rats and the moderate intensity exercise reduced the triglyceride levels to normal levels. Several animal and human studies have shown that serum triglycerides are negatively associated with endothelial function (see the Review by Zicha et al. Physiol. Res. 2006, 55 Suppl 1). Thus, it might be possible to speculate that triglycerides somehow affect the alterations of the vascular function. I suggest the authors discuss this in the Discussion section. 

3.      It is a pity that the authors did not provide the data on the expression SKCa and IKCa from mesenteric arteries of each group of rats. I was wondering why the data were not provided and why the authors did not refer to this point. 

Minor comments

1.      Discussion, line 520: The references authors cited here are mostly from the results from streptozotocin-induced rats, which is a type 1 diabetes mellitus model. I suggest the authors to cite the paper by Oniki et al. (Oniki et al. J Hypertens. 2006) which showed reduced EDH in mesenteric arteries from Goto-Kakizaki rats, a model of type 2 diabetes mellitus. 

2.      Discussion, line 546: Could the authors please cite appropriate references which showed “reduced anti-inflammatory effects of EDH”? 

3.      Discussion, lines 572-582: A recent interesting paper by Leo et al. (Leo et al. Am J Physiol Heart Circ Physiol 2021) demonstrated an upregulation of TMEM16A, a calcium-activated chloride channel, in resistance sized arterial smooth muscle cells of type 2 diabetes mice. Such a mechanism might also underpin the increased myogenic tone in mesenteric arteries of UCD-T2DM rats. Please discuss about it.

Round 2

Reviewer 2 Report

The authors have adequately addressed most of the concerns/suggestions and the manuscript is significantly improved. Still. the authors have not included the reference by Oniki et al. (Oniki et al. J Hypertens. 2006) which showed reduced EDH in mesenteric arteries from Goto-Kakizaki rats, a model of type 2 diabetes mellitus. The authors need to add this reference in the revised version so that to clarify that EDH is impaired not only in type 1 diabetes model but also in type 2 diabetes model.

Author Response

We apologize for missing the reference that was suggested by the reviewer. We have now added the study conducted by Oniki et al., 2006 which showed reduced EDH in mesenteric arteries from Goto-Kakizaki rats, a model of type 2 diabetes mellitus. Please see reference #70 in the Revised Manuscript.